# Visualization of Myocardial Strain Pattern Uniqueness with Respect to Activation Time and Contractility: A Computational Study

**Borut Kirn**

Medical Faculty, University of Ljubljana, Vrazov trg 2, 1000 Ljubljana, Slovenia; borut.kirn@mf.uni-lj.si;
Tel.: +386-1-543-75-00; Fax: +386-1-543-7501

**Abstract:** Speckle tracking echography is used to measure myocardial strain patterns in order to assess the state of myocardial tissue. Because electro-mechanical coupling in myocardial tissue is complex and nonlinear, and because of the measurement errors the uniqueness of strain patterns is questionable. In this study, the uniqueness of strain patterns was visualized in order to revel characteristics that may improve their interpretation. A computational model of sarcomere mechanics was used to generate a database of 1681 strain patterns, each simulated with a different set of sarcomere parameters: time of activation (TA) and contractility (Con). TA and Con ranged from −100 ms to 100 ms and 2% to 202% in 41 steps respectively, thus forming a two-dimensional 41 × 41 parameter space. Uniqueness of the strain pattern was assessed by using a cohort of similar strain patterns defined by a measurement error. The cohort members were then visualized in the parameter space. Each cohort formed one connected component (or blob) in the parameter space; however, large differences in the shape, size, and eccentricity of the blobs were found for different regions in the parameter space. The blobs were elongated along the TA direction (±50 ms) when contractility was low, and along the Con direction (±50%) when contractility was high. The uniqueness of the strain patterns can be assessed and visualized in the parameter space. The strain patterns in the studied database are not degenerated because a cohort of similar strain patterns forms only one connected blob in the parameter space. However, the elongation of the blobs means that estimations of TA when contractility is low and of Con when contractility is high have high uncertainty.

**Keywords:** signal uniqueness; visualization; myocardial strain; tissue properties; computational modeling; synthetic data

## 1. Introduction

Speckle tracking echography (STE) is regularly measured in cardiology [1–3]. It is used to obtain mechanics of the left ventricular wall as strain in different locations during a cardiac cycle thus generating a strain pattern. The contemporary analysis of the strain patterns is based on the prominent points found in the pattern like, peak strain value, time to peak strain rate, time to onset of shortening. They can be mapped with respect to the location in the wall or can be clustered as global ventricular value [4–6].

However, the information is stored in the entire strain pattern rather than only in the prominent points. To analyze it, a patient-specific modeling approach [7–11] is suggested. In patient-specific modeling, existing knowledge of physics and physiology is integrated within a computational model, which then serves to find the best model representation of in vivo measurements [12–15]. In the myocardial strain patterns, Le Rolle et al. [16] developed a suitable computational model and used it to find the model parameters by finding the best fit for simulated and measured local myocardial

strain patterns. As a result, they reported maps of model parameters that were corresponding to the myocardial tissue mechanical and electrical properties.

However, finding the best model representation of measured values by the fitting procedure is not useful when the model is complex, nonlinear [17–19], and the measurement errors are large; and in STE measurements the error margins are substantial [20,21]. In general, the drawback is that there could be more than one solution to fitting, and these solutions could even have drastically different parameter sets causing system degeneracy [22–24]. Furthermore, the sensitivity i.e., precision and accuracy of the obtained parameter values, might not be the same throughout the parameter space. Both characteristics would influence the interpretation of measured strains patterns.

To address the problem, we simplified the ventricular contraction and assumed that the mechanics of the sarcomere of each segment in the left ventricular wall depends on two basic tissue parameters: the contractility (Con) and time of activation (TA). This simplification enables simulation of the left ventricles with pathologies like left bundle branch block (LBBB) where timing of activation is altered, ventricles with myocardial infarction where contractility is altered, and combination of the two [25]. In this study, for all different combinations of tissue parameters TA and Con a strain pattern is simulated. By studying relations between the simulated strain patterns, their uniqueness is revealed.

The goal of this study is to visualize strain pattern uniqueness and thus to reveal the influence of system degeneracy and varying sensitivity on interpretation of STE measurements.

## 2. Methods

An open source lumped parameter computational model (CircAdapt) of cardiac mechanics and cardiovascular hemodynamics, which has integrated sarcomere mechanics , was used to simulate strain patterns in the left ventricular wall [26,27]. The sarcomere mechanics is based on a modified three-element Hill model (Figure 1A) stretched with given external tension *G(t)* (Figure 1B).

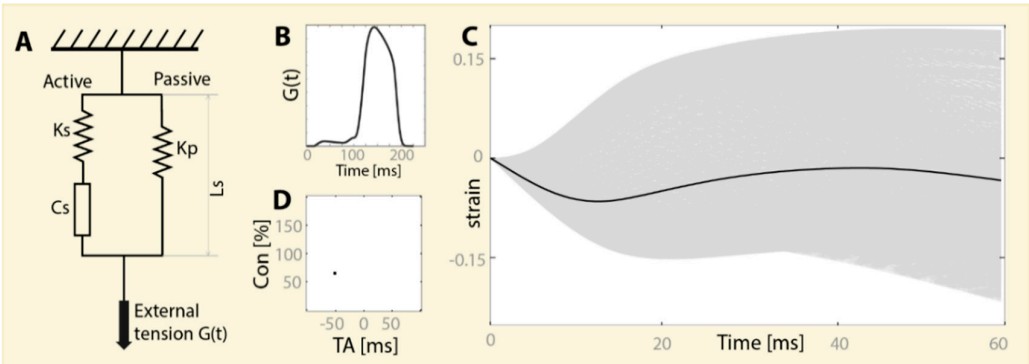

**Figure 1.** (**A**) Schematic presentation of a modified three-element Hill-based model of sarcomere mechanics. The "Active" branch is composed of a passive elastic element ($K_s$) and a contractile element ($C_s$); the "Passive" branch has a passive elastic ($K_p$) element. The length ($L_s$) of the sarcomere element is simulated for a given time of activation (TA) and contractility (Con) of the contractile element $C_s$. (**B**) External tension G(t) as a function of time was same for all simulations. G(t) was obtained from a heart failure patient simulation of a CircAdapt closed-loop cardiovascular model [26,27]. (**C**) Synthetic measurement space includes all 1681 simulated strains. The temporal resolution of the simulation was 2 ms. (**D**) Parameter space where one TA and Con pair is marked, the same as the one marked black in (C).

The model is written in Matlab, used ode24 to solve a system of differential equations, and takes about 1 s to calculate cardiac cycle on a standard personal computer. In total, 1681 simulations were calculated to obtain different strain patterns, each with a different set of sarcomere parameters TA and Con (Figure 1C). This database of strain patterns was then studied.

### 2.1. Parameter Space

The parameter space was set up as a grid of 41 × 41 fields of TA and Con pairs (Figure 1D). The TA ranged from −100 ms to 100 ms (41 values with 5 ms steps), and the Con parameter ranged from 2% to 202% (41 values with 5% steps). The TA was the time difference between the activation of the contractile element and the onset of external force. The Con parameter in the simulation was a scaling factor of the contractile element maximum tension.

### 2.2. Synthetic Measurement Space

The synthetic measurement space consisted of 1681 simulated strain patterns i.e., myofiber strain $E_f$ (Figure 1C) values were calculated from the sarcomere length $L_s$, which was a result of the model simulation, as

$$E_f(t) = \frac{L_s(t)}{L_{s,0}} - 1 \tag{1}$$

where $L_{s,0}$ denotes the sarcomere length at the time of closure of the mitral valve.

### 2.3. Visualization of the Cohort in the Parameter Space

For each simulated strain pattern, a cohort of similar strain patterns was found. Then an image of the cohort members was created in the parameter space, such that the position of each cohort member in the parameter space was marked. The condition for a cohort membership was that the mean distance between the strains was below the measurement error. The mean distance ($D$) was calculated as

$$D_{r,i} = \sqrt{\frac{\sum_{t=1}^{N}(E_{r,t} - E_{i,t})^2}{N}} \tag{2}$$

where $i$ stands for a cohort member candidate and $r$ for the reference strain pattern; $N$ runs from mitral valve closure until 2/3 into the duration of systole, which ended with the mitral valve opening. If $D_{r,i} \leq 0.03$, then the strain $i$ became a member of a cohort of reference strain $r$. The images of the cohort members for strain $r$ in the parameter space were visualized to determine their shape, size, and eccentricity.

## 3. Results

All 1681 simulated strain patterns are shown in Figure 1C. The cohorts of twelve reference strain patterns with distinctively different TA and Con values are shown in Figure 2A. These same cohort members are also presented in the parameter space (Figure 2B). The position of the reference strain is marked with a red square in the parameter space and with a red line in the synthetic measurement space. All 1681 cohorts in the synthetic measurement space and in the parameter space are shown in the appended video.

The simulated strain patterns in Figure 2A, subfigures 4, 5, 6 and 5, 8, 11 show characteristics of LBBB and myocardial infarction, respectively. In subfigure 4, TA is −50 ms, and the consequence of this early activation is initial fast shortening because the rest of the ventricle is not yet active. In subfigure 6, TA is 50 ms and the consequence of this late activation is initial fast stretching because the rest of the ventricle is already contracting. In subfigures 5, 8, and 11 Con is reduced from 100%, 25%, and 10%, respectively, and over time the strain becomes more positive with declining contractility, signifying a bulge in the ventricular wall.

The basic findings in the parameter space are two:

- Each image is composed of one blob (connected component);
- Each blob has a distinctively different shape, size, and eccentricity.

The shape of the blob in the region of higher contractility (Figure 2B, subplots 1–3) is stretched in vertical direction, spreading over up to 100% in the direction of the Con axes and 35 ms in the direction

of the TA axes. In the region of low contractility (Figure 2B, 7–12), the blobs are stretched in horizontal direction, covering up to 100 ms in TA and 20% in Con. In the regions of higher contractility and late activation (Figure 2B, subplots 3, 6, 9), the shape of the blob rotates from vertical toward horizontal when Con is reduced from 200% to 50%.

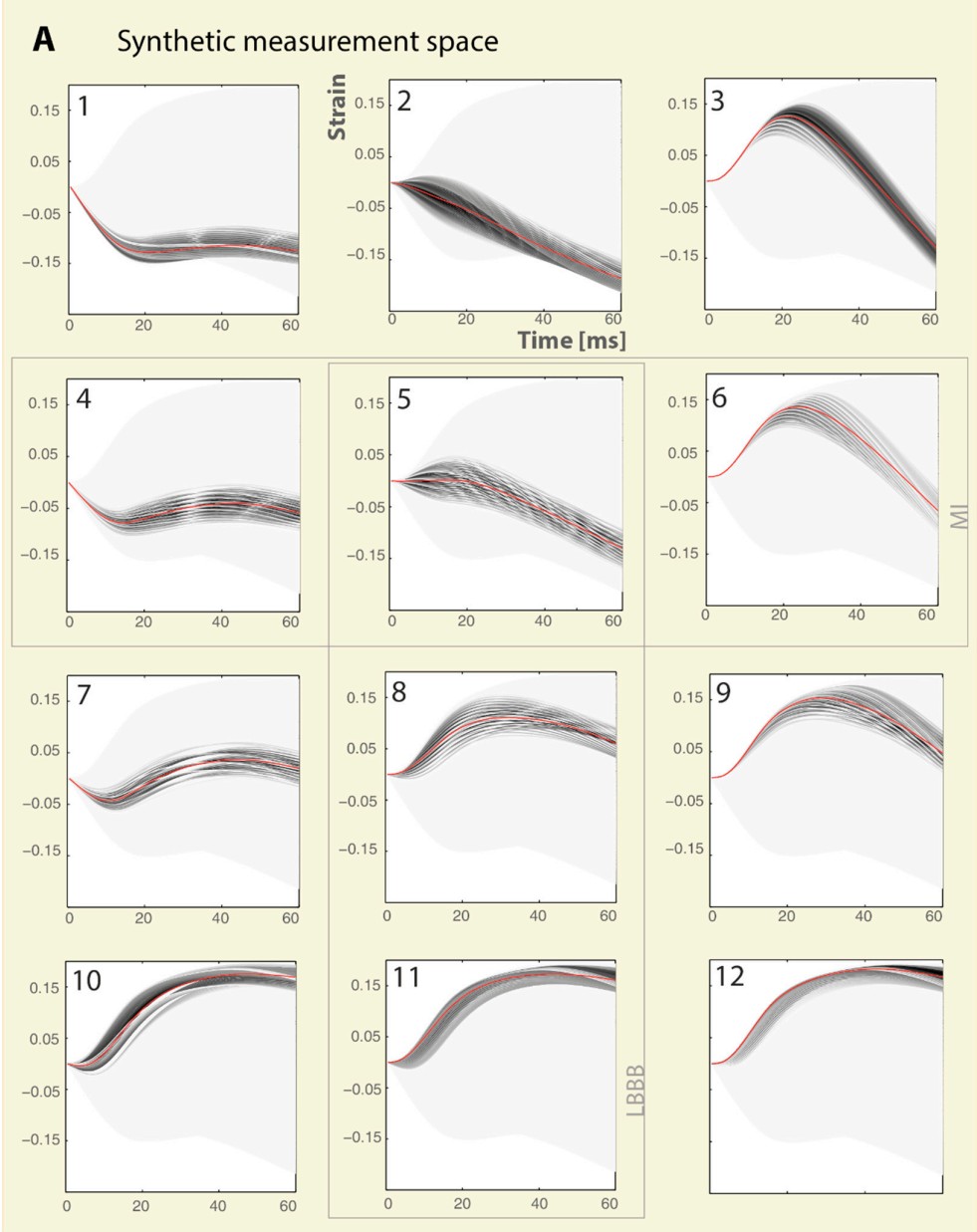

**Figure 2.** *Cont.*

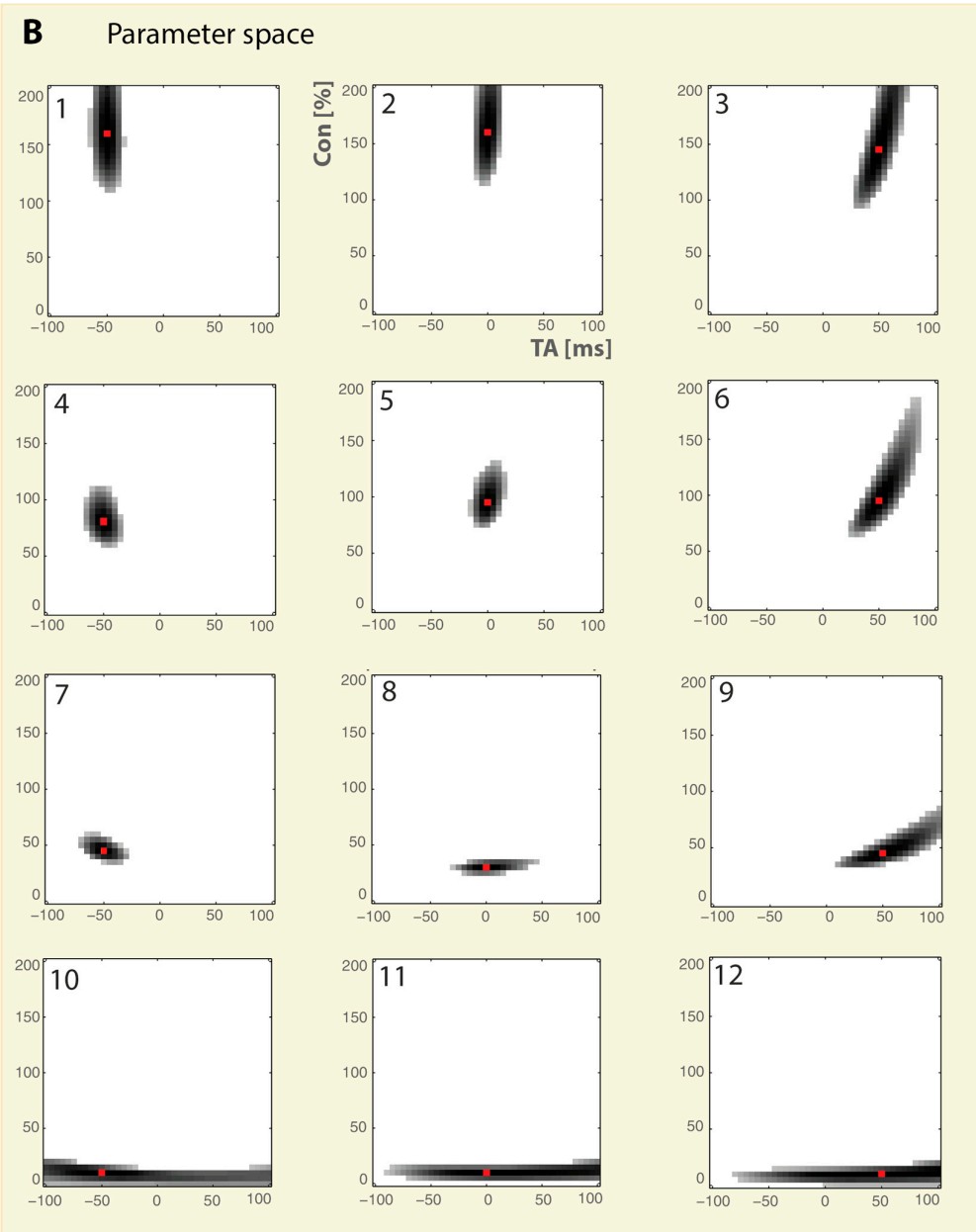

**Figure 2.** (**A**) Cohorts of 12 reference strains (1–12) with distinctively different TA and Con values; gray lines: cohort members, the intensity of gray falls with increased mean distance ($D_{r,i}$) from the reference strain (red); light gray lines: all simulated strain patterns. The strain patterns in subimages 4, 5, 6 and 5, 8, 11 are in accordance with the observed strain patterns in patients with left bundle branch block (LBBB) and myocardial infarction (MI), respectively. (**B**) The cohort members of reference strains 1–12 from A form one blob (connected component) in the parameter space. The intensity of gray falls with increased mean distance ($D_{r,i}$). The TA and Con values of the reference strain are marked with a red square.

## 4. Discussion

In this study, the uniqueness of the strain patterns was analyzed by a cohort of similar strain patterns visualized in the parameter space. No system degeneracy was found as the cohorts of all strain patterns formed only one blob. However, the sensitivity of the strain patterns on TA and Con depends considerably on the TA and Con values themselves, because the shape, size, and eccentricity of the blobs are drastically different in different regions in the parameter space.

Using a cohort of similar strain patterns is a kind of statistics where measurement error is translated into uncertainty in the synthetic measurement space, which is then reflected in the parameter space. In the parameter space, the blob can be further analyzed by means of shape and eccentricity. However, for the parameter extraction purpose the unique shapes of the blobs can be further used in developing TA and Con extraction techniques.

Changing of the blob shape within the parameter space is related to changing sensitivity of the strain patterns on TA and Con parameters. With respect to TA and Con parameters extraction, the blob's elongation translates into reduced precision. Whereas, an increase in the eccentricity i.e., non-centric position of the reference strain within the blob, means reduction in the accuracy of TA and Con extraction. Large blob elongations were found in the TA direction when contractility was low, and in Con direction when contractility was high. Thus, when interpreting STE measurements, clinicians should be aware that when contractility is low, estimation of TA values is unreliable and when contractility is high, estimation of Con values is unreliable.

The visualization technique developed transforms a complex biological signal into graphic problem and can thus be used universally to analyze signal uniqueness in a complex system. If the influence of more than two parameters on the signal is studied, then either multidimensional blobs would need to be analyzed or the dimensions of the parameter space would need to be reduced [28].

**Funding:** This research received no external funding.

**Conflicts of Interest:** The author declares no conflict of interest.

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
