# Peer review of "Visualization of Myocardial Strain Pattern Uniqueness with Respect to Activation Time and Contractility: A Computational Study"

_data, 2019_

Round 1

Reviewer 1 Report

This paper developed a computational model to simulate sarcomere mechanics. I have the following concerns: 1. The language usage throughout this paper need to be improved, the author should do some proofreading on it. Give the article a mild language revision to get rid of few complex sentences that hinder readability and eradicate typo errors. 2. What is the motivation of the proposed work? Research gaps, objectives of the proposed work should be clearly justified. The authors should consider more recent research done in the field of their study. OpenCMISS: a multi-physics & multi-scale computational infrastructure for the VPH/Physiome project in Progress in biophysics and molecular biology 107 (1), 32-47 3. The major trends of the simulation results may be shown in the bullet form. 4. Literature review techniques has to be strengthened by including the issues in the current system and how the author proposes to overcome the same. Discussion of related works in the strain should attract more readers: Value of three-dimensional strain parameters for predicting left ventricular remodeling after ST-elevation myocardial infarction The International Journal of Cardiovascular Imaging 33 (5), 663-673

Author Response

Dear Reviewer,

Thank you for meaningful comments which helped me to improve the manuscript towards exposing the aim and integration of the study into scientific community and hence addressing larger potential audience. Large section in the manuscript were rewritten (marked with red colour) particularly in Introduction and Discussing chapters. Thus, this point-by-point response includes conceptual explanations what has been changed. Please refer to submitted revised manuscript to apprehend the magnitude of manuscript changes.

Kind regards.

This paper developed a computational model to simulate sarcomere mechanics. I have the following concerns:

1. The language usage throughout this paper need to be improved, the author should do some proofreading on it. Give the article a mild language revision to get rid of few complex sentences that hinder readability and eradicate typo errors.

Large sections of the manuscript were rewritten in order to improve understandability of the manuscript.

2. What is the motivation of the proposed work?

Research gaps, objectives of the proposed work should be clearly justified.

The authors should consider more recent research done in the field of their study. OpenCMISS: a multi-physics & multi-scale computational infrastructure for the VPH/Physiome project in Progress in biophysics and molecular biology 107 (1), 32-47

To address points 1) and 2) a complete Introduction chapter was rewritten with large improvements in Bibliography by addition of 19 relevant references including the OpenCMISS study. The purpose of the study was exposed, the existing studies were named, and missing knowledge was exposed. Accordingly, parts of the Abstract were rewritten (first and last paragraph). The manuscript has now clearly explained aim, practical purpose and explanation how this technique is universally applicable.

3. The major trends of the simulation results may be shown in the bullet form.

Two major simulation results are now presented in bullet form.

4. Literature review techniques has to be strengthened by including the issues in the current system and how the author proposes to overcome the same.

Discussion of related works in the strain should attract more readers: Value of three-dimensional strain parameters for predicting left ventricular remodeling after ST-elevation myocardial infarction The International Journal of Cardiovascular Imaging 33 (5), 663-673 

The Literature review has been rewritten and large improvements in Bibliography were integrated including the mentioned study: Value of three-dimensional strain parameters for predicting left ventricular remodeling after ST-elevation myocardial infarction published in The International Journal of Cardiovascular Imaging

Reviewer 2 Report

The author shows that stain pattern uniqueness can be assessed and visualized. The idea is very interesting. However, the paper must be improved. The article is poorly organized, so the authors should improve it.

Main issues:

1)In the abstract and summary, there is not even one mention of the purpose of this paper. What is the meaning of visualization and assessment for practical purposes? Where else can it be used?

2)The bibliography is very poor, hence the introduction introduces little to the topic. The author should strongly expand the introduction and add the related works section.

3)The author should cite figures using the \ref{} commands to labels by a figure environment (\label{} should be after \caption{}) - if the autor used LaTeX format.

4)Description of coefficients and iterators should be written in a mathematical font, eg: iterator i or N in eq. (2).

5)Where is the experimental section?

6)Where is the statistical analysis?

7)What database was used in the research?

8)In what language or software has the method been implemented?

9)What is the computational complexity?

10)What is the time complexity?

Author Response

Dear Reviewer,

Thank you for meaningful comments which helped me to improve the manuscript towards exposing the aim and integration of the study into scientific community and hence addressing larger potential audience. Large section in the manuscript were rewritten (marked with red colour) particularly in Introduction and Discussing chapters. Thus, this point-by-point response includes conceptual explanations what has been changed. Please refer to submitted revised manuscript to apprehend the magnitude of manuscript changes.

Kind regards.

The author shows that stain pattern uniqueness can be assessed and visualized. The idea is very interesting. However, the paper must be improved. The article is poorly organized, so the authors should improve it.

Main issues:

1)In the abstract and summary, there is not even one mention of the purpose of this paper.

What is the meaning of visualization and assessment for practical purposes?

Where else can it be used?

2)The bibliography is very poor, hence the introduction introduces little to the topic. The author should strongly expand the introduction and add the related works section.

To address points 1) and 2) a complete Introduction chapter was rewritten with large improvements in Bibliography by addition of 19 relevant references. The purpose of the study was exposed, the existing studies were named, and missing knowledge was exposed. Accordingly, parts of the Abstract were rewritten (first and last paragraph). The manuscript has now clearly explained aim, practical purpose and explanation how this technique is universally applicable.

3)The author should cite figures using the \ref{} commands to labels by a figure environment (\label{} should be after \caption{}) - if the autor used LaTeX format.

            Not applicable because the manuscript is written in Microsoft Word.

4)Description of coefficients and iterators should be written in a mathematical font, eg: iterator i or N in eq. (2).

The Microsoft Word 2016 equation editor is used for equation. The fonts used for i, N in eq.2 are Cambria Math.

5)Where is the experimental section?

The experimental section is creating a database by using a computational model. In this study no measurements on subjects was used, only the results of computational simulations. To improve this explanation the first paragraph in the Methods was rewritten.

6)Where is the statistical analysis?

The approach used in the study namely creation of a cohort of similar strain patterns rather than taking one on its own is a kind of statistics. Because the study has no two different clinical measurements to be compared, the classical use of statistical t-tests is not relevant. To communicate the statistical aspect of this study to the readers a new paragraph was added into the Discussion.

Using a cohort of similar strain patterns is a kind of statistics where measurement error is translated into uncertainty in the synthetic measurement space which is then reflected in parameter space. In the parameter space, the blob can be further analyzed by means of shape and eccentricity however for the parameter extraction purpose the unique shapes of the blobs can be further used in developing TA and Con extraction technique.

7)What database was used in the research?

8)In what language or software has the method been implemented?

9)What is the computational complexity?

10)What is the time complexity?

In response to points 5 and 7-10, the first paragraph in the Methods chapter was complete rewritten. All technical, experimental and computational information requested by the reviewer are included.

An open source lumped parameter computational model of cardiac mechanics and cardiovascular hemodynamics which has integrated sarcomere mechanics (CircAdapt, circadapt.com, written in Matlab) was used to simulate strain patterns in the left ventricular wall [26, 27]. The sarcomere mechanics basis on modified three-element Hill model (Figure 1A) stretched with given external tension G(t) (Figure 1B). The model is written in Matlab, used ode24 to solve a system of differential equations and takes about 1s to calculate cardiac cycle on standard personal computer. In total 1,681 simulations were calculated to obtain different strain patterns each with a different set of sarcomere parameters TA and Con (Figure 1C). This database of strain patterns was then studied.

Round 2

Reviewer 1 Report

no more questions

Reviewer 2 Report

The authors have included most of my amendments. They justified the lack of an experimental section.